# LEARNING PROPOSALS FOR SEQUENTIAL IMPORTANCE SAMPLERS USING REINFORCED VARIATIONAL INFERENCE

**Zafarali Ahmed**
Mila - McGill University
zafarali.ahmed@mail.mcgill.ca

**Arjun Karuvally**
University of Massachusetts, Amherst
akaruvally@cs.umass.edu

**Doina Precup**
Mila - McGill University
dprecup@cs.mcgill.ca

**Simon Gravel**
Department of Human Genetics, McGill University
Genome Quebec Innovation Centre
simon.gravel@mcgill.ca

## ABSTRACT

The problem of inferring unobserved values in a partially observed trajectory from a stochastic process can be considered as a structured prediction problem. Traditionally inference is conducted using heuristic-based Monte Carlo methods. This work considers learning heuristics by leveraging a connection between policy optimization reinforcement learning and approximate inference. In particular, we learn proposal distributions used in importance samplers by casting it as a variational inference problem. We then rewrite the variational lower bound as a policy optimization problem similar to Weber et al. (2015) allowing us to transfer techniques from reinforcement learning. We apply this technique to a simple stochastic process as a proof-of-concept and show that while it is viable, it will require more engineering effort to scale inference for rare observations[1].

## 1 INTRODUCTION

Stochastic processes are an important model of sequential data in the natural sciences (Allen, 2010). Learning to fit theoretical models or sampling new sequential data that is consistent with the observed data is crucial for a deeper mechanistic understanding. Given that data from a stochastic process is ordered[2] in a trajectory, predicting the values of unobserved variables in a partially observed trajectory can be considered as a structured prediction problem.

For example, disease progression during an outbreak can be modeled by considering how the number of healthy, sick and recovered individuals changes every day (Allen, 2008). Given observations from an epidemic where we counted individuals towards the end of the outbreak, could we infer how the disease progressed at the start of the outbreak?

Approximating these distributions directly from data can be difficult and multiple methods based on heuristics have been proposed. In particular, many techniques use handcrafted Monte Carlo methods based on importance samplers, Markov Chain Monte Carlo or rejection sampling schemes (Gilks et al., 1995; Geyer & Thompson, 1995; Jiao et al., 2014; Sharma, 2017; Nelson et al., 2018). The promise of *automatically* learning heuristics from *raw* and *complex* data is one of the tantalizing goals of machine learning: It might allow us to use Monte Carlo methods for problems in which heuristics are not obvious.

Variational inference (VI, Blei et al. (2006; 2017)) is an alternative technique to approximate probability distributions that has seen increasing success when combined with deep learning (Kingma & Welling, 2013; Goodfellow et al., 2016; Gulrajani et al., 2016; Miao et al., 2016) in high dimensional

---

[1]Code is available at https://github.com/zafarali/better-sampling
[2]Usually the ordering variable is time.

datasets. Our work adds to existing techniques that enhance Monte Carlo methods using variational inference (Wexler & Geiger, 2007; Salimans et al., 2015; Naesseth et al., 2017; Müller et al., 2018).

Specifically, we consider the sampling of trajectories from the posterior distribution of a stochastic process using importance samplers. We formulate learning of proposal distributions as a VI problem. We then leverage the connection between VI and policy optimization reinforcement learning (Sutton et al., 2000; Bachman & Precup, 2015; Weber et al., 2015) to learn these proposal distributions. We apply *reinforced variational inference* (RVI, Weber et al. (2015)) to a simplified stochastic process as a proof-of-concept where we aim to sample trajectories, $\tau$, from a posterior over trajectories, $p(\tau|x_T)$, given a single terminal observation, $x_T$. Our contributions are:

1. Empirical validation of the RVI framework (Weber et al., 2015) suggests that RL is a viable approach for VI and more advanced tools from policy optimization might carry over to VI.

2. Learning of proposal distributions in sequential importance samplers can result in performance that is close to a hand-crafted importance sampler for most observed data. Qualitative evaluation demonstrates the reasonability of proposals found but quantitative evaluation identifies room for improvement.

## 2 BACKGROUND

In this section we describe technical background information regarding importance sampling, variational inference and reinforcement learning. These notions will be crucial to derive a variational approach to importance sampling, trained using policy optimization.

### 2.1 IMPORTANCE SAMPLING

The expectation of function $f$ under a distribution $p(\tau|x_T)$ can be estimated by drawing a large number of samples or *particles*, $\tau_i \sim p(\tau|x_T)$ and computing the sample mean: $\mathbb{E}_{p(\tau|x_T)}[f(\tau)] = \int p(\tau|x_T)f(\tau)\mathrm{d}\tau \approx \frac{1}{N}\sum_i f(\tau_i)$. In cases where $p(\tau|x_T)$ is difficult to sample from or the integral requires evaluating rare samples, we can rewrite the integral and expectation by drawing instead from a proposal distribution, $\tau_i \sim q(\tau)$, that is easy to sample from:

$$\int \frac{p(\tau|x_T)}{q(\tau)}q(\tau)f(\tau)\mathrm{d}\tau = \mathbb{E}_q\left[\frac{p(\tau|x_T)}{q(\tau)}f(\tau)\right] \approx \frac{1}{N}\sum_i \frac{p(\tau_i|x_T)}{q(\tau_i)}f(\tau_i). \tag{1}$$

This approach is known as *importance sampling*. We can think of the *importance weight*, $w_i = \frac{p(\tau|x_T)}{q(\tau)}$, as a way to correct for the relative probability of drawing $\tau$ according to $q(\tau)$ instead of $p(\tau|x_T)$.

Though Equation 1 is valid for any distribution $q$, the speed and quality of the approximation depends on $q$. In general, designing a $q(\tau)$ that is better than random is difficult but can be done for some simple posteriors (See Section S.1.2).

### 2.2 VARIATIONAL INFERENCE

Variational inference (VI) is an alternate approach to sample from a difficult distribution, $p(\tau)$, by sampling from a learnable approximate distribution $q(\tau; \theta)$ instead. The variational parameters $\theta$ are learned such that they minimize a divergence metric like the Kullback-Leibler or $\chi^2$ divergence with the true posterior: $q^*(\tau) = \arg\min_\theta D(q(\tau;\theta)||p(\tau|x_T))$. This procedure effectively turns the inference problem into a optimization problem (Blei et al., 2017; Dieng et al., 2017).

### 2.3 REINFORCEMENT LEARNING AND POLICY GRADIENT METHODS

In reinforcement learning (RL) (Sutton & Barto, 1998), we search for a policy $\pi$ that dictates how an agent should interact with the environment in order to maximize the cumulative reward. *Policy-based methods* (Sutton et al., 2000) involve parameterizing the policy $\pi_\theta(a|s)$ and then directly

optimizing: $\mathcal{O}_{ER} = \mathbb{E}_{\tau \sim \pi_\theta}[\sum_{t=0}^{T} \gamma^t r_t]$ where $r_t$ is the reward obtained at time $t$ by executing $\pi_\theta$ in the environment and $\gamma$ is a discount factor that prioritizes rewards to be sooner rather than later.

The gradient of $\mathcal{O}_{ER}$ can be approximated using the REINFORCE (Williams, 1992) estimator:

$$\nabla_\theta \mathcal{O}_{ER}(\theta) \approx \sum_{s_t, a_t \sim \pi_\theta} \nabla_\theta \log \pi_\theta(a_t|s_t) \Psi_t \tag{2}$$

where $\Psi_t$ is a target that describes how good the sampled action, $a_t$, was on the long term reward. We use the generalized advantage estimate (GAE) (Schulman et al., 2015b) as the target: $\Psi_t = \sum_{l=0}^{\infty} (\gamma\lambda)^l \delta_{t+l}$ where $\delta_{t+l} = r_t + \gamma V^\pi(s_{t+1}) - V^\pi(s_t)$ is the temporal difference residual (Sutton & Barto, 1998) and $V^\pi$ is a parameterized value function that estimates the long term discounted reward. We also experiment with an *entropy regularized* version of this algorithm where the entropy of the policy, $\mathbb{H}(\pi)$, is added to the objective to prevent learned policies from becoming deterministic too quickly (Schulman et al., 2017a).

## 3 APPROACH USING REINFORCED VARIATIONAL INFERENCE (RVI)

We begin by casting the problem of learning a proposal distribution as a variational inference problem in Section 3.1. We then draw the connection with policy optimization in Section 3.2 which results in a similar formulation to Weber et al. (2015).

### 3.1 THE VARIATIONAL INFERENCE PROBLEM

Recall that we are interested in sampling from the posterior over trajectories, $p(\tau|x_T)$ given a single terminal observation, $x_T$. We consider Markov processes where the recursive relationship $x_t = x_{t-1} + \Delta x_t$ holds between random variables. Only a fixed number of transitions, $\Delta x_t$, are possible and we model this as a *discrete* random variable with probability distribution $p(\Delta x|x_{t-1})$. The probability of observing a trajectory from the stochastic process $\tau = (x_0, \dots, x_T)$ can be decomposed as a product of the transitions: $p(\tau) = p(x_0) \prod_{t=0}^{T-1} p(\Delta x_t|x_{t-1})$.

Recall that variational inference introduces a simpler approximating posterior. Let $q(\overleftarrow{\tau}|x_T)$ be the approximate posterior over trajectories that can be written as a product of local parameterized conditionals called *proposals*: $q(\overleftarrow{\tau}|x_T) = \prod_{t=0}^{T-1} q_\theta(\Delta x_t|x_{t+1})$.

Since we are given the ground-truth terminal observation, $x_T$, the trajectory is created by going *backwards* in time starting the sampling from $q_\theta(\Delta x_{T-1}|x_T)$, computing the next state $x_{T-1}$ and repeatedly sampling from $q_\theta(\Delta x_{t-1}|x_t)$ until reaching $t = 0$. We can construct the trajectory by using $x_t = x_{t+1} - \Delta x_t$ where $\Delta x_t$ is one of the outcomes possible under the forward transition probabilities. The KLD cannot be minimized exactly, and instead we maximize the evidence lower bound (Blei et al., 2017):

$$D_{KL}(q(\overleftarrow{\tau}|x_T)||p(\tau|x_T)) \leq - \underbrace{\left[ \mathbb{E}_q[\log p(\tau)] - \mathbb{E}_q[\log q(\overleftarrow{\tau}|x_T)] \right]}_{\text{Evidence lower bound (ELBO)}} \tag{3}$$

In the next section, we will see how to transform the maximization of the ELBO into a policy optimization problem and draw the connection with rewards in reinforcement learning.

### 3.2 THE CONNECTION TO POLICY OPTIMIZATION

To demonstrate the link with RL, we substitute the full forms of the variational distribution, $q(\overleftarrow{\tau}|x_T)$ and trajectory probability, $p(\tau)$ into the ELBO (Equation 3):

$$\mathbb{E}_{q_\theta}\left[ \log \frac{p(\tau)}{q(\overleftarrow{\tau}|x_T)} \right] = \mathbb{E}_{q_\theta}\left[ \log \frac{p(x_0) \prod_t^{T-1} p(\Delta x_t|x_t)}{\prod_t^{T-1} q_\theta(\Delta x_t|x_{t+1})} \right] \tag{4}$$

$$= \mathbb{E}_{q_\theta}\left[ \log p(x_0) + \sum_{t=0}^{T-1} \log \frac{p(\Delta x_t|x_t)}{q_\theta(\Delta x_t|x_{t+1})} \right] \tag{5}$$

By comparing with REINFORCE (Equation 2), Weber et al. (2015) view this as a RL problem where at state $x_{t+1}$ the policy $q$ takes an action $\Delta x_t$ in an environment and receives a reward

$r_t = \log \frac{p(\Delta x_t | x_t)}{q_\theta(\Delta x_t | x_{t+1})}$ and moves to state $x_t$. In the final step, we also obtain $r_f = \log p(x_0)$ as a reward. The reward has an appealing interpretation: It is the relative log likelihood of executing the action $\Delta x_t$ in the stochastic process versus that of the policy. It encourages the policy to choose actions that would be likely under the generative stochastic process. We write Equation 5 as $\mathbb{E}_{q_\theta}[\sum_t r_t + r_f]$, to show the equivalence to the RL objective with no discounting[3], i.e. $\gamma = 1$. Equation 5 suggests that we can use off-the-shelf reinforcement learning algorithms to optimize the parameters of the proposal distribution $q(\tau)$ by learning the one step proposal $q_\theta(\Delta x_t | x_{t+1})$ as the policy.

An algorithm to obtain a trajectory from proposal and subsequently update it would proceed in three phases:

1. **Collect a trajectory:** Sample $\Delta x_t \sim q_\theta(\Delta_t | x_t)$ and update the state $x_{t-1} = x_t - \Delta x_t$. Now $x_{t-1}$ is used to obtain the next sample and is repeated until $t = 0$.

2. **Update the proposal:** For each $\Delta x_t$ that is obtained, rewards are calculated as $r_t = \max(R_{\min}, \log \frac{p(\Delta x_t | x_t)}{q_\theta(\Delta x_t | x_{t+1})})$ and $r_f = \log p(x_0)$ and used to compute the empirical return[4]. The gradient is then estimated using REINFORCE and a GAE target and used to update the parameters via stochastic optimization.

3. **Update the posterior:** If the final position $x_0$ is valid under the prior, i.e. $p(x_0) > 0$, then the trajectory is saved with the weight $w_i = p(x_0) \prod_t \frac{p(\Delta x_t | x_t)}{q_\theta(\Delta x_t | x_{t+1})}$.

## 3.3 USING THE $\chi^2$ DIVERGENCE

In this section, we will motivate the $\chi^2$ divergence as alternative to the KLD. We then demonstrate the flexibility of the RVI framework by extending it to minimize this divergence and making a connection to sparse reward reinforcement learning.

Assume that we want to estimate the likelihood of an observation $x_T$, which requires taking an intractable integral over all possible paths: $p(x_T) = \int p(x_T, \tau) d\tau$. We can use importance sampling with a parameterized proposal distribution $q_\theta(\tau)$ so that we can estimate $p(x_T)$ with fewer samples and lower variance. Specifically,

$$p(x_T) = \int p(\tau, x_T) d\tau \approx \frac{1}{N} \sum_{n=0}^{N-1} w(\tau) = \hat{p}(x_T) \qquad (6)$$

where $\tau \sim q_\theta(\cdot)$ and $w_\tau = \frac{p(x_T, \tau)}{q_\theta(\tau)}$ are the importance weights. Burda et al. (2015) used such an importance sampling estimate of the log-likelihood to improve variational autoencoders. One property we might desire from the importance sampling proposal is that it minimizes the variance of $\hat{p}(x_T)$:

$$V(\hat{p}(x_T)) = \frac{1}{N} \left\{ E_q[w(\tau)^2] - E_q[w(\tau)]^2 \right\} \qquad (7)$$

$$= \frac{1}{N} \left\{ E_q \left[ \left( \frac{p(\tau)}{q(\tau)} \right)^2 \right] - \hat{p}(x_t)^2 \right\} \qquad (8)$$

which corresponds to minimizing the $\chi^2$-divergence: $D_{\chi^2}(p(\tau) || q(\tau)) = \mathbb{E}_{q_\theta} \left[ \left( \frac{p(\tau)}{q(\tau)} \right)^2 - 1 \right]$.

In Dieng et al. (2017), the likelihood ratio estimate for the gradient is derived for the exponentiated "$\chi^2$ Upper Bound" or CUBO, $\frac{1}{2} \log E_q[\left( \frac{p(\tau)}{q(\tau)} \right)^2]$, which is an upper bound on $D_{\chi^2}(p(\tau) || q(\tau))$:

$$\nabla_\theta \exp \left( 2 \frac{1}{2} \log E_q \left[ \left( \frac{p(\tau)}{q(\tau)} \right)^2 \right] \right) \approx -\frac{1}{N} \sum_n \left( \frac{p(\tau^{(n)})}{q(\tau^{(n)})} \right)^2 \nabla_\theta \log q(\tau^{(n)}) \qquad (9)$$

---

[3]In practice we still use discounting close to 1 for optimization stability.

[4]$R_{\min}$ is used here to avoid issues where we might have $-\infty$ rewards.

Unlike the KLD, the $\chi^2$-divergence cannot be broken down into a summation. Instead, we can treat $-\left(\frac{p(\tau)}{q(\tau)}\right)^2$ as a sparse reward that only appears at the end of a trajectory. Since we still build trajectories by sampling from the local conditionals $q_\theta(\Delta x_t|x_{t+1})$, we can never assign credit to any *particular* action. Sparse reward problems are generally quite difficult as there are no intermediate signals to guide learning (Ng et al., 1999). Our work will only evaluate the feasibility of replacing the KLD with the $\chi^2$ and we will leave investigation of densification to future work.

## 4 RELATED WORK

Many efforts have been made to combine variational optimization and Monte Carlo methods. We cover a few approaches here and, where relevant, describe connections and deviations with existing literature. See Appendix S.2 for a wider discussion on the connections between policy optimization, variational inference and importance sampling.

We note that the formulation provided here was also provided in Weber et al. (2015) for more generic stochastic computation graphs. Our work grounds this framework in a concrete approximate inference problem.

Grover et al. (2018) used rejection sampling to minimize the KL divergence using a *resampled* ELBO: Proposed samples are rejected if they have a low likelihood under the model resulting in a *resampled* proposal distribution. This is particularly similar to our work as their accept-reject step is *differentiable* which means it can be adapted over time.

The most similar to our work is Gu et al. (2015) and Naesseth et al. (2017) who learned proposal distributions in the sequential Monte Carlo setting. Our work deviates in one crucial way: we learn discrete transitions between states in the sequence. This allows some increased flexibility at the cost of being potentially high variance due to the gradient estimator. Indeed Naesseth et al. (2017) use the reparameterization technique (Kingma & Welling, 2013) that provides a low variance gradient estimator for continuous distributions[5]. In particular, Naesseth et al. (2017) derive a differentiable filtering algorithm that combines variational optimization and an expectation-maximization algorithm to automatically learn both the proposal and forward model.

Recently Müller et al. (2018) used variational inference by optimizing for proposals represented by normalizing flows (Rezende & Mohamed, 2015) to learn sampling distributions for constructing *light paths*. Like this work, the authors optimize the $\chi^2$ divergence and obtained similar qualitative results.

Markov Chain Monte Carlo (MCMC) methods are another family of techniques to approximate probability distributions. Salimans et al. (2015) used variational optimization to learn the transition operator of the Markov chain that is used in MCMC algorithms. Though this kind of algorithm can be extended to the sequential setting it would require sampling many times from the Markov Chain. Our work avoids the use of a transition operator for the Markov chain by directly requiring the proposal distribution to produce a valid transition within the stochastic process itself.

## 5 RESULTS

In this section we will describe some quantitative and qualitative empirical results that show the viability of the problem formulation and approach using RVI. We will also highlight failure modes of the method.

### 5.1 EXPERIMENTAL SETUP

**Environments:** We consider stochastic processes based on random walks as environments (Section S.1). They are chosen specifically to demonstrate the usefulness of the technique and not because they are particularly complex. Random walks form the basis for more complex processes.

---

[5]While reparameterization techniques exist for discrete distributions, they are not differentiable without making a continuous relaxation (Jang et al., 2016; Maddison et al., 2016).

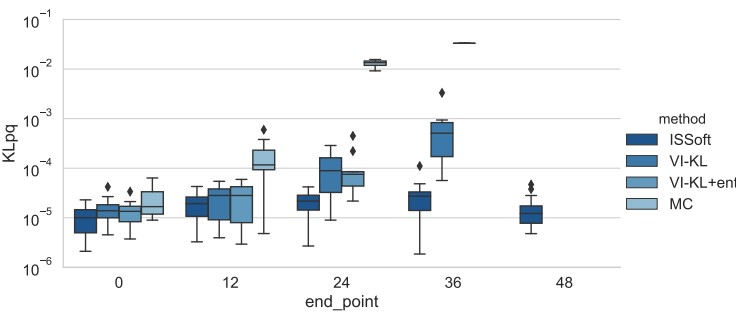

Figure 1: **KL divergence (KLpq) between the approximate and true posterior for different samplers organized by the starting distance from the origin,** $x_T \in \{0, 12, 24, 36, 48\}$, **after 250,000 samples.** As the distance increases, methods like naive Monte Carlo (MC) begin to fail. Using RVI-KLD can learn to perform as well as the hand crafted importance sampler when the $x_T$ is close to the window and where MC cannot perform at all. RVI begins to fail for more extreme cases, $x_T \in \{36, 48\}$.

Specifically, we use the simple 1D random walk whose initial point is described by a discrete uniform distribution with a window (-5, 5) and evolves by taking a step +1 or -1 with equal probability for $T = 50$ timesteps (See complete details in Section S.1.1). We control the ending point, $x_T$, to modulate the *difficulty* of the posterior estimation problem, i.e. as $x_T \to 50$ the observation becomes rarer and the posterior over starting positions becomes skewed towards values near the edges of the window. The proposal, $q_\theta$, is a categorical distribution over possible transitions, $\Delta x_t$.

**Quantitative Evaluations:** To measure the online performance of importance samplers, we report the KL Divergence between the approximated and true posterior[6] over starting positions (KLpq). Note that the KLpq is only *one* downstream metric of interest and presents some limitations. To avoid overfitting to one metric, we also consider qualitative behaviours to better understand the different components of the learned proposals and the posteriors.

**Qualitative Evaluations:** To understand the adaptive behaviour of the proposal, we observe how trajectories sampled from the proposal change during the optimization procedure. We investigate the reasonability of proposal learned by visualizing the expected direction a particle will move at any point $x_t$ at any time $t$. Finally, we use a histogram to compare the approximated posterior over starting positions to the true posterior to understand the sensitivity in KLpq.

**Baselines:** We consider two baselines: (1) Naive Monte Carlo (MC) where the transition probability $p(x_t|x_{t-1})$ is used as the proposal $q(x_{t-1}|x_t)$; and (2) Hand-crafted importance sampler with a soft proposal (ISSoft) where the backward probability is biased toward the window depending on the average number of steps needed to be within the range of the window[7].

## 5.2  QUANTITATIVE PERFORMANCE

We first investigate the viability of learning proposals online that are specific to each ending observation. Unsurprisingly, ISSoft performs well for all ending observations and is therefore a simple but strong baseline to beat. After sampling and training with 250,000 trajectories RVI-KL performs as well as ISSoft for observations that are close to the window $x_T \in \{0, 12\}$. For observations that are far away from the window, $x_T \in \{24, 36\}$, RVI-KL performs better than the random MC method but not as good as the hand-crafted ISSoft (Figure 1).

## 5.3  QUALITATIVE ANALYSIS

In this section we take a look at qualitative aspects of learning proposals as well as final posteriors learned. To understand if the posterior samples better trajectories after successive steps of optimization, we measure the proportion of trajectories that end within the window during optimization. Compared to the start of the optimization, a larger number of trajectories sampled from the proposal end within the window (Figure S1).

---

[6]For the random walk, the posterior over starting positions can be derived exactly. See Section 18.

[7]The probabilities are given by Equation 20. See Section S.1.1 for a complete definition.

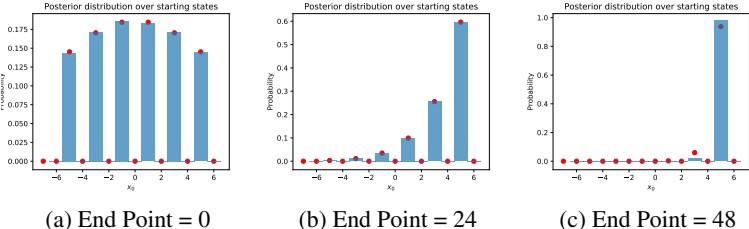

(a) End Point = 0        (b) End Point = 24        (c) End Point = 48

Figure 2: **Approximated posterior distribution over starting states when using KLD rewards.** There is a high degree of matching between the approximated (blue bars) and true distributions (red dots). For $x_T = 48$, many posteriors do not estimate the low probability values in the support well which explains the high KLpq. Note that even values have zero probability of occurring since starting between (-5, 5) and continuing for 50 steps cannot result in an even number.

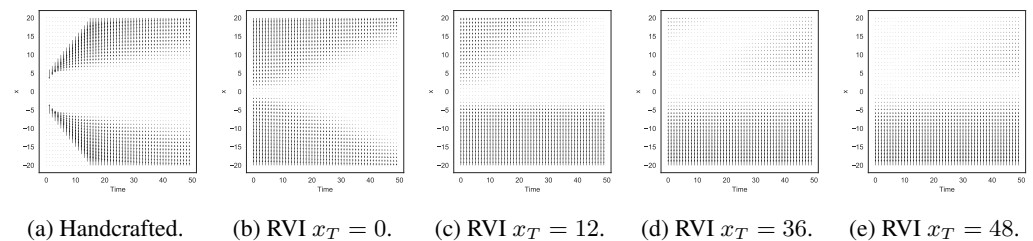

(a) Handcrafted.    (b) RVI $x_T = 0$.    (c) RVI $x_T = 12$.    (d) RVI $x_T = 36$.    (e) RVI $x_T = 48$.

Figure 3: **A variety of proposals are learned conditioned on the ending observation, $x_T$.** (a) The hand-crafted proposal. (b-e) The learned proposals after 250,000 trajectories. When the observation, $x_T$, is far from the window of starting positions, the proposal learned is skewed to push trajectories toward the window. Arrows show the expected step at any given $x$ and $t$ combination.

To understand the poor quantitative performance (Figure 1), we visualize the posterior over starting positions, $p(x_0|x_T)$, obtained by RVI-KL (Figure 2) compared to the true analytics posterior. For ending points that are close to the window, the posterior recovered is near-perfect for both 1D (Figure 2a and 2b) and 2D random walks (Figure S2). Even though we were unable to record quantitative performance for $x_T = 48$ (Figure 1), the posteriors found captured the general shape, and only missed the low probability mass values of $x_0$ (Figure 2c).

Visualizing the proposal reveals sufficient specialization for each ending observation (Figure 3). Particularly interesting is the progressively skewed proposals obtained when the ending point changes from $x_T = 0$ to $x_T = 36$. Indeed the proposal learned when $x_T = 48$ is not completely unreasonable as it prevents the particle from moving into the region where $x_i < 0$.

| | | | end_point, $x_T$ | | |
| method | 0 | 12 | 24 | 36 | 48 |
|---|---|---|---|---|---|
| ISSoft | **0.000033** | **0.000039** | **0.000058** | **0.000068** | **0.000036** |
| RVI-KL+pre+eval | 0.000034[†] | **0.000055** | **0.000055[†]** | **0.000207** | NaN |
| RVI-KL+pre+fine | **0.000034[†]** | 0.000108 | 0.003550 | NaN | NaN |
| RVI-KL | 0.000030[†] | 0.000118 | 0.000720 | 0.005527 | NaN |
| RVI-KL+ent | **0.000031[†]** | **0.000093** | **0.000661** | 0.056438 | NaN |
| RVI-C2 | 0.004475 | 0.011863 | 0.026869 | **0.002469** | **0.000806** |

Table 1: **Mean performance for different objectives and training schemes after 100,000 trajectories** Bold font represents best performing methods in each category. † represents better-than or on par with ISSoft as measured by overlap in the errors. See full table with standard errors in Table 2.

## 5.4 An alternate objective function

In this section, we devise an experiment to probe whether we are using the right objective function. In particular, we *pre-train* a proposal by using the algorithm outlined in Section 3.2 and changing the ending observation, $x_T$, after each trajectory (Figure S3a)[8]. Evaluation of the pre-trained proposal by measuring KLpq after 100,000 trajectories reveals that it performs on-par with ISSoft and better than training from scratch: The model is of sufficient representational capacity and the pre-training scheme can produce a proposal that generalizes well to different end points (Table 1, RVI-KL+pre+eval). Next, we repeat the experiment but this time allowing the proposal to be *fine-tuned* (Figure S3b and S3c). Surprisingly, performance becomes worse compared to no fine-tuning at all: Even though the initialization is good, the training scheme resulted in worse performance. One possible reason could be the use of the wrong objective function (Table 1, RVI-KL+pre+fine).

In Section 3.3 we showed how minimizing the $\chi^2$ divergence is equivalent to minimizing the variance of the estimator of the log-likeihood. Furthermore, we showed how it could be used in the RVI framework. We find that this technique does not perform well (Table 1, RVI-C2) compared to using KLD (Table 1, RVI-KL). Despite not having good quantitative performance, the correct shape of the approximate posterior has been captured (Figure S4).

## 6 Discussion and Future Work

Our experiments have shown that policy optimization can be a viable approach to learn interesting proposals that, in some instances, performs on par with hand-crafted ones. This result shows us the viability of two things: (1) Framing sequential variational inference problems as policy optimization problems; (2) Learning proposals that can approximate posteriors for stochastic processes. These observations are encouraging and suggest that more advanced techniques from policy optimization, like generalized advantage estimation (Schulman et al., 2015b) or asynchronous advantage actor critic (Mnih et al., 2016), can be brought to variational inference[9]. Therefore, RVI can learn proposals for *simple* stochastic processes. Despite the approach being easily extended to different stochastic processes given the forward dynamics, it might require significant future work to be able to learn posterior distributions for more advanced stochastic processes.

We identify at least two areas for improvement. In particular, poor results on rare instances of $x_T$ that are far from the window highlights the need for better data sampling strategies[10]. Secondly, it is somewhat paradoxical that RVI cannot learn to improve upon pre-trained proposals: The local optimum obtained by training on the distribution of ending points, i.e. pre-trained, is worse than if we trained from scratch or specialized the pre-trained distribution. This means that the KLD objective function being optimized is not reflective of the kind of downstream tasks we want to evaluate the model on. One way to check this would be to directly optimize for the downstream metric and verify the quality of the KLD objective and will be left for future work. The latter part of this work considered alternative objectives such as the $\chi^2$ divergence, but is inconclusive as to its merits.

One possible solution to make transfer more feasible is to impose a penalty that ensures that proposals do not deviate from the pre-trained proposal unless there is a significant benefit and is a known approach in policy optimization (Schulman et al., 2015a). Perhaps the gradients are too noisy to find a better optimum and a better multi-sample gradient estimator (Mnih & Rezende, 2016) or variance reduction techniques should be used.

In this work we have investigated and empirically validated that we can leverage connections between policy optimization and variational inference to learn proposals in importance sampling. Our results are encouraging but also highlight the amount of engineering needed to get these methods to work for more difficult problems.

---

[8]Full details in Section S.3.1.

[9]We observed progressively improved optimization by using more sophisticated targets, $\Phi_t$.

[10]Commonly referred to as exploration in reinforcement learning.

ACKNOWLEDGMENTS

The authors would like to thank Riashat Islam for detailed feedback on this work as well as the helpful comments from anonymous workshop reviewers. This work relied on funding from the Canada Graduate Scholarship program via the Canadian Institutes of Health Research and Google Summer of Code. Computational resources were provided from Compute Canada and its regional partners.

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

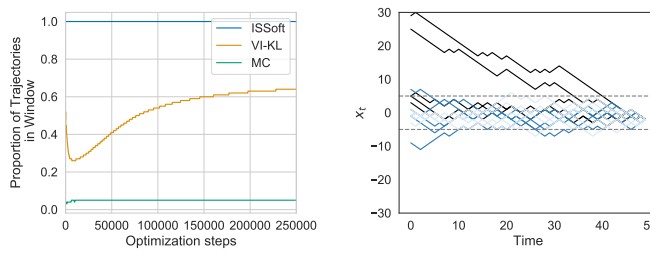

(a) Prop. of successful trajectories.     (b) Sampled trajectories

Figure S1: **Evidence for adapting proposals** (a) The proportion of trajectories that ended in the window increases as optimization progresses; (b) Example trajectories sampled during the optimization end within the window (dashed lines). Dark colors represent trajectories sampled during the first 100 optimization steps. Light colors represent trajectories sampled during the last 100 optimization steps.

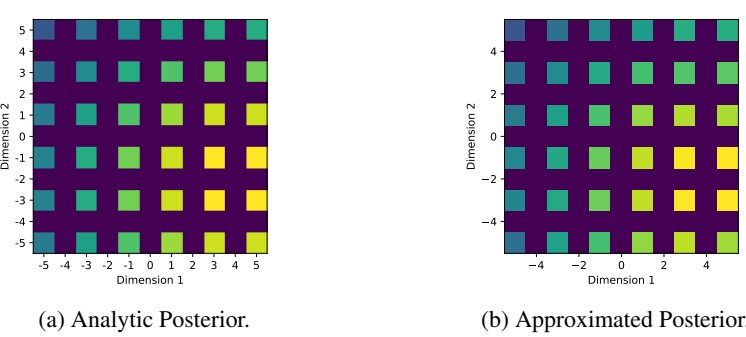

(a) Analytic Posterior.        (b) Approximated Posterior.

Figure S2: **Exact and approximated posteriors over starting locations for the two dimensional random walk given the ending observation point is** $x_T = \{4, -2\}$**.** The agreement is near-perfect.

# S    SUPPLEMENTARY MATERIAL

## S.1    RANDOM WALK

The random walk is a simple stochastic process (Pearson, 1905) where the sequential dependence between two variables, $X_t$ and $X_{t+1}$, in a random walk is described by:

$$X_{t+1} = X_t + \Delta X \tag{10}$$

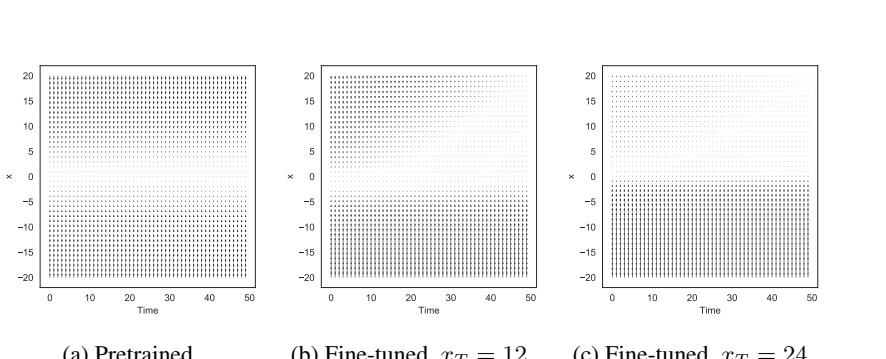

(a) Pretrained.   (b) Fine-tuned, $x_T = 12$.   (c) Fine-tuned, $x_T = 24$.

Figure S3: **Proposals obtained by pre-training and then fine-tuning.** (a) Proposal obtained by training with multiple $x_T$; (b-c) Specialization of the pretrained proposal based on the last observation.

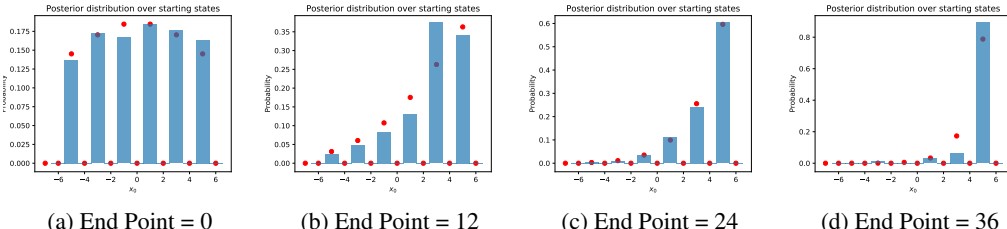

|  |  |  |  |
|---|---|---|---|
| (a) End Point = 0 | (b) End Point = 12 | (c) End Point = 24 | (d) End Point = 36 |

Figure S4: **Approximated posterior distribution over starting states when using C2 rewards.** There is a degree of mismatch between the true posterior (red dots) and the approximated posterior (blue bars), however, the overall shape has been captured.

where the transition, $\Delta X$, is drawn according to the following probability distribution:

$$P(\Delta X | X_t) = \begin{cases} p & \text{if } \Delta X = +1 \\ 1 - p & \text{if } \Delta X = -1 \end{cases} \tag{11}$$

The first random variable is drawn from a discrete uniform prior, $P(X_0 = x) = \frac{1}{|\mathcal{X}|} \; \forall x \in \mathcal{X}$, where $|\mathcal{X}|$ is the size of the support. Though this may seem like a trivial process, it has widespread use in financial mathematics, modelling animal movement, infectious diseases, biophysics and even computer science.

### S.1.1 DERIVING THE ANALYTIC POSTERIOR FOR THE RANDOM WALK

Though sampling from the posterior over trajectories, $p(\tau | x_T)$, can be difficult for arbitrary stochastic processes, in the case of the simple random walk described in Section S.1, the posterior over starting states, $p(x_0 | x_T)$, can be calculated exactly. We use this result to compute a downstream measure for the performance of the approximate distribution obtained by different methods considered in the experimental section.

For the purposes of clarity, we will override notation for this section: Let $S_t = \Delta X_t$ as defined in Equation 11; $X_0$ the random variable describing the starting position; and $D = X_T$ the random variable describing the final position.

Our goal is to derive the posterior over starting states, $p(X_0 \mid D)$. Using Bayes Rule:

$$p_{X_0 | D}(X_0 \mid D) = \frac{p_{D | X_0}(D \mid X_0) p_{X_0}(X_0)}{p_D(D)} \tag{12}$$

Therefore, we can first derive $p_{D | X_0}(D | X_0)$ and then apply Bayes rule to obtain the desired posterior distribution.

| method | end_point, $x_T$ | | | | |
|---|---|---|---|---|---|
|  | 0 | 12 | 24 | 36 | 48 |
| ISSoft | $0.000033 \pm 0.000004$ | $0.000039 \pm 0.000005$ | $0.000058 \pm 0.000009$ | $0.000068 \pm 0.000008$ | $0.000036 \pm 0.000008$ |
| RVI-KL+pre+eval | $0.000034 \pm 0.000005$ | $0.000055 \pm 0.000009$ | $0.000055 \pm 0.000018$ | $0.000207 \pm 0.000078$ | NaN |
| RVI-KL+pre+fine | $0.000034 \pm 0.000007$ | $0.000108 \pm 0.000034$ | $0.003550 \pm 0.003092$ | NaN | NaN |
| RVI-KL | $0.000030 \pm 0.000007$ | $0.000118 \pm 0.000035$ | $0.000720 \pm 0.000224$ | $0.005527 \pm 0.002945$ | NaN |
| RVI-KL+ent | $0.000031 \pm 0.000006$ | $0.000093 \pm 0.000031$ | $0.000661 \pm 0.000200$ | $0.056438 \pm NaN$ | NaN |
| RVI-C2 | $0.004475 \pm 0.001486$ | $0.011863 \pm 0.004135$ | $0.026869 \pm 0.010958$ | $0.002469 \pm 0.001587$ | $0.000806 \pm 0.000311$ |

Table 2: **Mean performance and standard error for different training schemes after 100,000 trajectories**

Since we are taking steps of size +1 or -1, the random variable describing our step, $S_t$, is a shifted Bernoulli, i.e. $S_t = 2X_t - 1$ where $X_t \sim \text{Bern}(p)$. Explicitly, $S_t$ has the following distribution:

$$p_{S_t}(S_t = s) = p_{X_t}\left(X_t = \frac{s+1}{2}\right) = \begin{cases} p & \text{if } \frac{s+1}{0} = 1 \\ 1-p & \text{if } \frac{s+1}{2} = 0 \end{cases} \tag{13}$$

If the walk is of length $T$, the distribution of the total displacement due to the steps taken, $S_T = \sum_{i=1}^{T} S_t$, is given by:

$$p_{S_T}(S_T = s) = p_{S_T}\left(\sum_{i=1}^{T} S_t = s\right)$$

$$= p_{S_T}\left(\sum_{i=1}^{T}(2X_t - 1) = s\right)$$

$$= p_{S_T}\left(\sum_{i=1}^{T} X_t = \frac{s+T}{2}\right) \tag{14}$$

We know that $\sum_{i=1}^{T} X_t$ is distributed according to the Bernoulli distribution. Therefore:

$$p_{S_T}(S_T = s) = p_{S_T}\left(\sum_{i=1}^{T} X_t = \frac{s+T}{2}\right)$$

$$= \binom{T}{\frac{s+T}{2}} p^{\frac{s+T}{2}}(1-p)^{T-\frac{s+T}{2}} \text{ if } \frac{s+T}{2} \in [0, T] \tag{15}$$

We now have all the terms necessary to compute the numerator of Equation 12. We first consider $p_{X_0}(X_0)$. Since it is drawn uniformly within a window between $-c$ and $+c$, i.e. $X_0 \sim \text{DiscreteUniform}(-c, c)$, the probability mass function is given by

$$p_{X_0}(x) = \left\{ \frac{1}{2c+1} \quad x \in [-c, c]; 0 \text{ otherwise} \right\} \tag{16}$$

Next, we focus on $p_{D|X_0}(D \mid X_0)$. Note that the total distance moved is $D = X_0 + S_T$. Therefore, we can write:

$$p_{D|X_0}(D = d \mid X_0 = x) \propto p_{D|X_0}(X_0 + S_T = d \mid X_0 = x)$$

$$= p_{S_T|X_0}(S_T = d - x \mid X_0 = x)$$

$$= p_{S_T}(S_T = d - x) \tag{17}$$

We then have that $P_{X_0|D}(d, x) \propto P_{S_T}(d - x)P_{X_0}(x)$. This lets us write the posterior as:

$$P_{X_0|D}(X_0 = x|D = d) \propto \frac{1}{2c+1}\binom{T}{\frac{d-x+T}{2}} p^{\frac{d-x+T}{2}}(1-p)^{T-\frac{d-x+T}{2}} \tag{18}$$

if $x \in [-c, c]$ and $\lfloor \frac{d-x+T}{2} \rfloor \in [0, T]$ and 0 otherwise. $\binom{a}{b} = \frac{a!}{(a-b)!b!}$ is the binomial coefficient.

### S.1.2 The difficulty of designing good proposals

Consider the simple example of sampling trajectories from the posterior of the random walk introduced in Section S.1. Since the random walk is unbiased[11], we could use a Naive MC method: construct a trajectory backward, $\overleftarrow{\tau}$[12], by considering a new walk starting from $x_T$. Specifically, we sample $x_t$ according to $q(x_t|x_{t+1}) = p(x_{t+1}|x_t)$.

The process described works remarkably well but poses a problem when $x_T$ has drifted *far* from the window of possible starting positions. Sampling backward according to the forward probabilities will likely result in the final sampled position, $\hat{x}_0$, being close to $x_T$[13].

---

[11]Behaviour is the same whether it is run forward or backward.

[12]We use $\overleftarrow{\tau}$ to refer to trajectories sampled backward such that $x_{t+1}$ was drawn before $x_t$. Similarly, $\overrightarrow{\tau}$ denotes trajectories that were sampled forward, such that $x_t$ was drawn before $x_{t+1}$.

[13]Since the walk is unbiased, it is not expected to deviate far from where it started.

To combat this, we can use an importance sampling scheme to sample from a $q$ that is different from $p$, but ensures that we obtain trajectories that are biased towards the window. This will result in $x_0$'s that are more likely under the prior and therefore a better estimate of the posterior. The question now becomes *How would we design a proposal $q$?*

For the problem of the random walk, designing a $q$ that is better than random is easy: Given that we know particles started in a window defined between $-c$ and $c$, we can design a proposal that pushes $x_t$ toward the window. We first calculate a bias term, $b$, which measures, on average, how many steps are needed to move toward the window at any given time $t$ during the walk. We use a softness coefficient, $s$, to define how strong the bias should be:

$$b = \begin{cases} \frac{(l-x_t)s}{(T-t)} & \text{if } x_t < l \\ \frac{(r-x_t)s}{(T-t)} & \text{if } x_t > r \\ 0 & \text{otherwise} \end{cases} \qquad (19)$$

where $T$ is the total length of the walk. If the particle at time $t$, $x_t$, drifts too far from the window, i.e. the average number of steps needed is larger than the step size , $|b| > 1$, it is not possible to return to a position where the window is in reach within a reasonable time frame and a random action is executed instead. Otherwise, with probability $p = 1 - |b|$ we take a random step and probability $1 - (1 - |b|)$ a step in the bias direction. Specifically, the probability of each action is:

$$q(\Delta X | x_t) = \begin{cases} \frac{p}{2} + (1-p)\mathbf{1}(b > 0) & \text{for } \Delta X = -1 \\ \frac{p}{2} + (1-p)\mathbf{1}(b < 0) & \text{for } \Delta X = +1 \end{cases} \qquad (20)$$

where $\mathbf{1}$ is the indicator function: It is 1 when the argument is true and 0 otherwise.

## S.2 Connections between importance sampling, variational inference and policy optimization

### Implications for importance sampling

In this work we have viewed sequential importance sampling as a variational inference problem. We demonstrated the flexibility of the approach in learning proposals to situations where they are not obvious. The implication of this result is encouraging: Applying importance sampling to non-traditional problems where proposals are not obvious. Given that VI works well with high dimensional real world sequential data (Bachman & Precup, 2015; Goyal et al., 2017; Müller et al., 2018) we conjecture that it can be used to learn complex proposal distributions in domains beyond vision and text.

Given that the sequential KLD breaks down into sums of rewards, we leveraged the fact that we did not need to optimize the joint probability of the whole trajectory: The transition at every step could be weighted by the action probability similar to the per-decision importance sampler in Precup (2000).

### Implications for variational inference

Our work has shown the viability of using policy optimization algorithms for sequential variational inference problems. This suggests that state-of-the-art methods from reinforcement learning can be brought into variational inference (Kakade, 2002; Schulman et al., 2015a; 2017b) for learning approximating distributions. For example, we transferred the concept of "$\lambda$-return" using generalized advantage estimation to sequential VI (Schulman et al., 2015b).

Beyond the obvious scheme of transferring methods, we can also consider more subtle concepts from reinforcement learning. For example, we can consider the KLD terms in the RVI framework as a *generalized value function* and combine these with auxiliary tasks (Sutton et al., 2011) that can stabilize learning. Concepts like eligibility traces for credit assignment might be able to allow us to use a more diverse set of reward specifications (Singh & Sutton, 1996).

### Implications for reinforcement learning

Though we have explored probabilistic inference as reinforcement learning, there has been a renewed interest in viewing reinforcement learning as probabilistic inference (Levine, 2018). All

these connections are under-explored and will be the subject of new research in the next few years. The KLD makes an appearance in a variety of policy optimization algorithms to regularize policy improvement steps. Perhaps some work should be put into exploring different divergences. For example, Neumann (2011) uses the reverse KLD to obtain policies with more cautious behaviour.

More directly related to this work is the concept of learning a backward model and was successfully introduced as *recall traces* in Goyal et al. (2019). Most recently, Piche et al. (2019) have successfully used a (learned) sequential Monte Carlo algorithm to learn promising future trajectories to plan in continuous control reinforcement learning problems. Significant research will surface in the next few years from exploring this connection more closely.

Policy optimization is a special case of gradient descent on an importance sampled loss function (Jie & Abbeel, 2010): When estimating importance sampled objectives, we could re-use trajectories collected at any time before a given update to estimate a gradient. Mnih & Rezende (2016) shows an approach to multi-sample REINFORCE-like objectives that may be useful in policy optimization when using multiple samples. The idea further generalizes by thinking of off-policy learning where we have a target policy, $\pi$, that we wish to learn and a behaviour policy $\mu$ from which data is collected. The forward model serves the same purpose as $\mu$ and the proposal of $\pi$. Revisiting these notions of off-policy policy optimization by using tools from variational inference is a promising direction (Fellows et al., 2018).

## S.3 IMPLEMENTATION DETAILS

We represent policy and value functions using two independent three-layer neural networks with RELU non-linearities (Glorot et al., 2011) implemented in PyTorch (Paszke et al., 2017). Each layer has 32 hidden units. We use the RMSProp update rule during optimization (Tieleman & Hinton, 2012). We make our code open source at https://github.com/zafarali/better-sampling.

### S.3.1 FINE-TUNING EXPERIMENT

To pre-train the proposal distribution we train $q$ on a range of final positions, $x_T$, obtained by sampling an initial point, $x_0 \sim p(x_0)$, and doing a rollout. This allows $q$ to be trained from the natural distribution of ending points.

To fine-tune the model, we take the pre-trained model from above and continue training with $x_T$ held fixed to the end point of interest.

