# OpenReview forum: "Learning proposals for sequential importance samplers using reinforced variational inference"
_ICLR.cc/2019/Workshop/drlStructPred — drlStructPred 2019_

### Official Review · AnonReviewer4 · 2019-04-06
**Preliminary investigation of RVI on toy task**

**Rating:** 3
**Confidence:** 3

**Review:**

This work investigates the connection between RL and variational inference on a toy task. Building upon the RVI  (Weber et al., 2015), off-the-shelf RL methods like policy gradient methods can be applied to the variational inference reformulated as RL. The authors attempted to apply this idea to learn a proposal distribution in an importance sampler. Based on RVI, they also proposed alternative objective that replaces the KL divergence with chi-square divergence. An inference problem in simple 1D random walk is used for evaluation. The proposed approach is shown to outperform simple MC baseline, but still worse than a hand-crafted baseline. And the objective using chi-square divergence isn't performing well.

The paper is relatively well-written and easy to follow. This work is an empirical investigation of RVI on a toy task, and proposes the use of chi-square divergence as an alternative to KL-divergence.

Although the experiment is quite preliminary and the novelty is limited, this is an interesting work in progress and I recommend acceptance. I would suggest exploring some real inference tasks in graphical model as additional benchmark to make the experiments more solid.

Minor issues:
Page 1 paragraph 4 "Variational inference (VI, Blei et al. (2006; 2017)) is an alternate technique" --> "alternative technique"
The equation (6) should have "x_T" instead of "x_t".

---

### Official Review · AnonReviewer3 · 2019-04-06
**Empirical performance of reinforced variational inference**

**Rating:** 4
**Confidence:** 2

**Review:**

This paper proposes an empirical validation of the reinforced variation inference method proposed by Weber et al. 2015, as a  good candidate for transferring existing policy optimization techniques from variational inference.
The results, however, are not consistent, this work reports reasonable results on qualitative experiments, but quantitative experiments suggest that this approach does not perform better than handcrafted proposals on experiments with random walks. Furthermore, experiments with Chi-squared showed that is not a better alternative than KL-divergence.

This paper provides preliminary results on an interesting topic but with inconclusive remarks, I believe further investigation should be done, in particular regarding the effect of performing RVI on real VI problems and comparing against different VI approaches not only random MC. It would be interesting to understand where does this approach fail, for which cases? It is also not clear why the finetuned approach fails when compared with the simple pretrained model. How different is the proposed approach from Weber 2015? I believe a remark in the related work would better help understand the differences.
Is RVI leveraging techniques from policy optimization, such as GAE and A3C, it would be interesting to see whether this is helping in the future.

This paper is written clearly and provides a thorough explanation of results, although some details could be clarified, such as more information on how the model is pretrained and fine tuned?  why would this objective be a wrong objective function? (5.4), and what are the family of proposal distributions  (q_theta) adopted.

I believe this paper provides

Minor revisions:
equation equation -> equation (2.1)
amd -> and (5.4)
aas -> as (FigureS1)

---

### Official Review · AnonReviewer5 · 2019-04-06
**Improvements in inferred unobserved values in partially observed trajectory**

**Rating:** 3
**Confidence:** 1

**Review:**

This paper is well written and describes their goals and reasons for their decision making clearly. They clearly state their objectives and use of variational inference in as a reinforcement learning problem to improve trajectories. They provide comparisons to Random Monte Carlo methods as well as ISSoft demonstrating improvements of MC methods and in some instances similar performance to ISSoft. The pros are their clear evidence of being able to use variational inference as a means of matching hand crafted baselines as well as a clear reinforcement learning objective to improve the quality of trajectories. The cons are that their proposed method can learn proposals only for a simple stochastic process and do note that they recieve poor results on rare instances from their X distribution as well as their approach not being able to improve the quality  of pre-trained proposals.

---

### Official Review · AnonReviewer1 · 2019-04-07
**interesting proof of concept paper**

**Rating:** 4
**Confidence:** 2

**Review:**

In this paper, the authors propose the use of REINFORCE for selecting proposals in importance sampling. The idea is to write  the problem as a variational inference task, and then establish the link between policy optimization and variational lower bound. The paper is relatively well-written, and it seems like it's above the bar of the workshop to me.

A few items:

1. The experiments are a bit weak, in the sense that the dataset and experimental settings descriptions are a bit vague, and it does not seem that the authors have tried real-world datasets such as vision, games, and language problems.

2. It's widely known that MCMC method could be very inefficient when sampling from complex distributions. That's why some people prefer likelihood-free methods such as neural variational inference. It would be great to add some discussions and comparisons. I would appreciate some analysis on runtime since REINFORCE is also bad at sample complexity.

3. I haven't seen much interesting theoretical results from this paper. Maybe you can ask more interesting theoretical questions? Such as the sample complexity? Regret bounds?

Anyway, I think it's a nice paper showing the connections among VI, REINFORCE, and importance sampling, but the theoretical and empirical results could be strengthened in many different ways.

---

### Decision · Program_Chairs · 2019-04-08
**Acceptance Decision**

Accept